



# A comparison of the momentum budget in reanalysis datasets during sudden stratospheric warming events

Patrick Martineau[1], Seok-Woo Son[2], Masakazu Taguchi[3], Amy H. Butler[4]

[1]Research Center for Advanced Science and Technology, The University of Tokyo, Tokyo, Japan
[2]School of Earth and Environmental Sciences, Seoul National University, Seoul, South Korea
[3]Department of Earth Science, Aichi University of Education, Kariya, Japan
[4]University of Colorado Cooperative Institute for Research in Environmental Sciences

*Correspondence to*: Patrick Martineau (pmartineau@atmos.rcast.u-tokyo.ac.jp)

**Abstract.** The agreement between reanalysis datasets, in terms of the zonal-mean momentum budget, is evaluated during sudden stratospheric warming (SSW) events. It is revealed that there is a good agreement among datasets in the lower stratosphere and troposphere concerning zonal-mean zonal wind, but less so in the upper stratosphere. Forcing terms of the momentum equation are also relatively similar in the lower atmosphere, but their uncertainties are typically larger than uncertainties of the zonal wind tendency. Similar to zonal wind tendency, the agreement among forcing terms is degraded in
the upper stratosphere. Discrepancies among reanalyses increase during the onset of SSW events, a period characterized by unusually large fluxes of planetary-scale waves from the troposphere to the stratosphere, and decrease substantially after the onset. While the largest uncertainties in the momentum budget originate from the Coriolis torque, momentum flux convergence also presents a non-negligible spread among the reanalyses. Such a spread is reduced in the latest reanalysis products, decreasing the uncertainty of the momentum budget. It is also found that the uncertainties of the momentum budget
depend on the strength of SSW events: the strongest SSWs being subject to larger discrepancies among reanalyses. These uncertainties in stratospheric circulation, however, are not communicated to the troposphere.

## 1 Introduction

Sudden Stratospheric Warming (SSW) events are prime manifestations of coupling between the tropospheric and
stratospheric circulations (Baldwin and Dunkerton, 2001). They are characterized by a rapid deceleration and reversal of the stratospheric zonal wind resulting from an enhanced injection of planetary-scale wave activity from the troposphere to the stratosphere (Limpasuvan et al., 2004; Martineau and Son, 2015; Polvani and Waugh, 2004). The changes in stratospheric circulation can then, in return, influence tropospheric weather (Kidston et al., 2015). Motivated by their role for tropospheric predictability (Charlton et al., 2004; Hardiman et al., 2011; Mukougawa et al., 2009; Sigmond et al., 2013; Taguchi, 2015;
Tripathi et al., 2015), SSWs have been the object of many observational and modelling studies.



Recent studies have highlighted a sensitivity to the choice of reanalysis dataset for the detection of SSW events. For instance, Charlton and Polvani (2007) noted discrepancies in the central date of SSW events between ERA-40 and NCEP-NCAR. They also found discrepancies in the classification of those events per the geometry of the distorted stratospheric polar vortex, whether it is displaced off the pole or split into two daughter vortices. Song and Chun (2016) also found some differences in ERA-Interim, JRA-55, and MERRA. Other definitions of SSW events also show discrepancies among reanalyses (Butler et al., 2015). Since SSWs are often defined using a threshold value, requiring a reversal of zonal-mean zonal wind ($\overline{u} = 0$) at 10 hPa and 60ºN (Charlton and Polvani, 2007), their detection can be sensitive to small differences in the zonal wind between reanalyses. Despite those discrepancies, Palmeiro et al. (2015) have noted that the general signature of SSW events is not sensitive to the choice of reanalysis. Furthermore, composites of the northern annular mode (NAM) index during a common set of SSW event dates for all reanalyses were shown to be similar (Martineau and Son, 2010).

Despite the seemingly good agreement of zonal-mean zonal-wind, temperature, and geopotential height between datasets during SSW events, inter-dataset variability merits further investigation. Lu et al. (2015) recently highlighted non-negligible differences in the wave drag and the residual circulation between ERA-40 and ERA-Interim. A comprehensive comparison of momentum diagnostics in the stratosphere among reanalyses further revealed non-negligible inconsistencies in the zonal-mean momentum equation, resulting primarily from inter-data variability in the residual circulation in the mid-stratosphere (Martineau et al., 2016). Such variability of the Brewer-Dobson circulation among reanalysis datasets is well documented in the literature (Abalos et al., 2015; Iwasaki et al., 2009; Monge-Sanz et al., 2013).

Martineau et al. (2016) have also shown that the ability to explain the stratospheric zonal-mean zonal wind tendencies using the forcing terms of the zonal-mean momentum equation has improved in the latest reanalysis products and that momentum diagnostics tend to agree better among the latest reanalyses. This improvement was demonstrated in the context of the wintertime climatology and different regimes of vortex variability (strong/weak, accelerating/decelerating) with an emphasis on the mid-stratosphere. In this work, we focus on the most extreme events of stratospheric variability, the SSW events, and extend the comparison of the momentum budget to the upper stratosphere and troposphere.

The reanalysis datasets evaluated in this work are first presented in Section 2.1, followed by a description of the momentum diagnostics used throughout this study in Section 2.2. The SSW events used for the comparison of reanalyses are then presented in Section 2.3. The uncertainty of stratospheric vortex structures among reanalysis datasets is first evaluated in Section 3 using as a case study the January 2009 SSW event. Then, zonal-mean quantities and diagnostics using the zonal-mean momentum equation are shown in Section 4. Conclusions are finally presented in Section 5.



## 2 Methodology

### 2.1 Data

The eight reanalysis datasets compared in this study are listed in Table 1. ERA-40 from the European Centre for Medium-Range Weather Forecasts (ECMWF) is excluded as it is not provided for recent years, thus limiting the sample size of SSW

events since 1980. The NOAA Twentieth Century Reanalysis (20CR) and the ECMWF Twentieth-Century Reanalysis (ERA-20C) are also left out as they are known to have unrealistic stratospheric variability in comparison to reanalyses that fully assimilate upper atmospheric observations (Compo et al., 2011; Poli et al., 2013). Temperature and the three-dimensional wind field are used on pressure levels for each reanalysis. To prevent our diagnostics from being affected by the vertical resolution, which gives an unfair advantage to the latest reanalyses, only 22 common vertical levels are kept (i.e.,

1000, 925, 850, 700, 600, 500, 400, 300, 250, 200, 150, 100, 70, 50, 30, 20, 10, 7, 5, 3, 2, and 1 hPa) except for NCEP-NCAR and NCEP-DOE which are available only up to 10 hPa. Similarly, to ensure that the diagnostics are not affected by differences in horizontal resolution, each dataset is interpolated onto a standardized 2.5° by 2.5° grid, which is the coarsest grid provided among all the reanalyses considered in this study. This reduction of the resolution in some reanalyses is not expected to have a large impact on our comparison. In fact, horizontal and vertical resolutions were previously shown not to

have a substantial effect on momentum diagnostics except near the tropopause and in the upper stratosphere where higher vertical and horizontal resolutions improved slightly the dynamical consistency (Martineau et al., 2016). The reanalysis datasets are compared for a common period ranging from 1980 to 2012, 1980 being the first year for which MERRA2 data is provided and 2012 being the official final year for the comparison of reanalyses in the SPARC Reanalysis Intercomparison Project (S-RIP) project. An introduction to S-RIP and a comprehensive description of the reanalyses are provided in

Fujiwara et al. (2017).

**Table 1: Summary of reanalysis datasets included in the comparison**

| NAME | Label | Highest level (hPa) | Original Resolution[$] | Reference |
|---|---|---|---|---|
| ERA-Interim[LRE] | E-I | 1 | 1.5 | Dee et al. (2011) |
| NCEP–NCAR Reanalysis-1* | N-N | 10 | 2.5 | Kalnay et al. (1996) |
| NCEP–DOE Reanalysis-2 | N-D | 10 | 2.5 | Kanamitsu et al. (2002) |
| NCEP Climate Forecast System Reanalysis[LRE,+] | N-C | 1 | 2.5 | Saha et al. (2010) and Saha et al. (2014) |
| JRA-25 | J25 | 1 | 2.5 | Onogi et al. (2007) |
| JRA-55[LRE] | J55 | 1 | 1.25 | Kobayashi et al. (2015) |
| MERRA | ME | 0.1 | 1.25 | Rienecker et al. (2011) |
| MERRA2[LRE] | ME2 | 0.1 | 1.25 | Gelaro et al. (2017) |

* Vertical velocity is not provided above 100 hPa



$^\$$ The original resolution is not necessarily the highest resolution provided by each reanalysis center.
$^{LRE}$ Included in the Latest Reanalysis Ensemble (LRE)
$^+$ Transition from version 1 to version 2 on January 1 2011

Whenever the average or standard deviation of multiple reanalyses is taken as a reference, it is performed on a subset including the latest reanalysis products from each center (ERA-Interim, NCEP-CFSR, JRA-55 and MERRA2). This composite is referred to as the latest reanalysis ensemble (LRE). In some figures, the LRE is contrasted to a composite of all reanalyses which we denote as the all reanalysis ensemble. The LRE subset emphasizes the discrepancies affecting the
reanalysis datasets that are nowadays most commonly used in research while excluding older reanalyses whose deficiencies are well documented in the literature.

**2.2 Momentum diagnostics**

The zonal-mean momentum equation, derived from the primitive version of the equation in pressure coordinate, is expressed as

$$\frac{\partial \overline{u}}{\partial t} = \underbrace{f\overline{v}}_{fv} \underbrace{- \frac{1}{a\cos^2\phi} \frac{\partial(\cos^2\phi \overline{u'v'})}{\partial\phi}}_{du'v'/dy} \underbrace{- \overline{v}\frac{1}{a\cos\phi}\frac{\partial(\overline{u}\cos\phi)}{\partial\phi}}_{Adv_\phi} \underbrace{- \overline{\omega}\frac{\partial\overline{u}}{\partial p}}_{Adv_p} \underbrace{- \frac{\partial(\overline{u'\omega'})}{\partial p}}_{du'\omega'/dp} + R \quad (1.1)$$

where $f$ is the Coriolis parameter, $u$, $v$, $\omega$ are the zonal, meridional, and vertical components of wind, $\phi$ is the latitude, and $p$ is the pressure. Overbars and primes denote zonal mean and anomalies with respect to the zonal mean, respectively. While the left-hand side term expresses the zonal-mean zonal wind tendency, terms of the right-hand side represent forcing terms. They are, in order, the acceleration due to the Coriolis torque, the meridional convergence of
momentum fluxes, the advection of zonal momentum by the meridional wind, the vertical advection of zonal momentum by the vertical wind, and the vertical convergence of vertical momentum fluxes. The last term, $R$, is referred to as the residual and represents sub-grid scale processes such as gravity wave drag and numerical diffusion. It also includes imbalances in the momentum equation introduced by the data assimilation process (analysis increment), errors due to the interpolation from model levels to pressure levels, and errors related to the numerical methods employed to evaluate each term of the equation.
R can be used to quantify and compare the consistency of the momentum budget among reanalysis datasets (Lu et al., 2015; Martineau et al., 2016; Smith and Lyjak, 1985). The quasigeostrophic (QG) version of the momentum equation is often applied in the extratropics. Under this approximation, only the first two terms on the right-hand side of Eq. (1.1) are retained. The validity of this approximation can be assessed by comparing the magnitude of non-QG terms ($3^{rd}$ to $5^{th}$ right hand side terms of Eq. (1.1)) to the QG terms. The numerical methods employed to evaluate Eq. (1.1) are described in further details in
Martineau et al. (2016). Abbreviations of the various terms of Eq. (1.1) used in figures throughout this paper are indicated with braces in Eq. (1.1). In this work, the Eulerian-mean form of the momentum equation is preferred over the transformed Eulerian mean since additional vertical derivatives are needed in the latter, which can introduce numerical errors. The



dynamical processes responsible for the eddy fluxes and the dominant terms of the momentum budget during SSW events are illustrated schematically in Fig. 1.

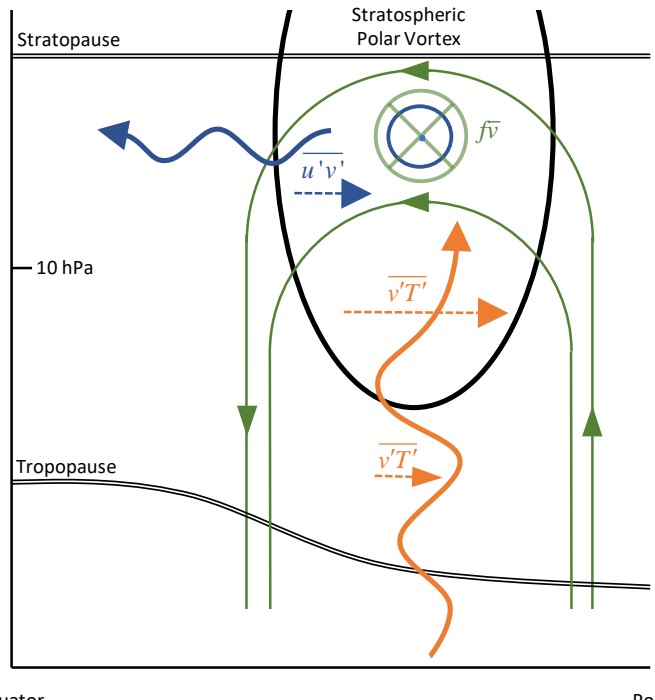

**Figure 1: Schematic illustration of the dominant forcing terms of the momentum budget in the Eulerian framework during SSW events and their underlying dynamical processes. Planetary-scales waves propagate upward (orange) from the troposphere to the stratospheric polar vortex (thick black line) and are accompanied with poleward heat fluxes ($\overline{v'T'}$) amplifying with height. In agreement with the Matsuno (1971) model, the poleward heat fluxes generate a thermally direct circulation (green) with air rising at the pole and sinking in the mid-latitudes. To close this circulation, equatorward motion is generated in the stratosphere. The Coriolis torque resulting from this Circulation decelerates the wind (into the page symbol). Equatorward wave propagation is also observed (blue), accompanied with poleward fluxes of westerly momentum ($\overline{u'v'}$). The resulting convergence of momentum fluxes in the polar stratosphere acts to accelerate the zonal-mean zonal wind (out of the page symbol), counteracting partly the deceleration by the Coriolis torque. Note that in the transformed Eulerian mean the residual circulation is opposite, i.e., poleward in the stratosphere and downward at the pole** (Song and Chun, 2016)**. The residual circulation approximates Lagrangian-mean motion.**





## 2.3 Definition of SSW events

SSW events are defined following Charlton and Polvani (2007). Their central dates are set as the dates when the zonal wind reverses direction ($\bar{u} = 0$) in winter at 10 hPa and 60ºN. Contrary to the original WMO definition, we do not verify if a reversal of the zonal-mean temperature gradient occurs at the same time as wind reversal. This additional criterion affects

5  minimally the outcome of the detection (Charlton and Polvani, 2007). As mentioned earlier, since the definition of SSW events is based on a threshold value, the detection of a SSW can be sensitive to small variations in the wind field. Because of large variations in the characteristics of individual SSW events (e.g., Ayarzaguena et al., 2011; Martineau and Son, 2013), the comparison of reanalysis datasets could be negatively affected by using a different set of events for each reanalysis. To ensure a fair comparison, SSWs are first detected in each reanalysis independently. If at least 4 reanalyses detect the SSW

10  event, the onset date is set by averaging across the dates given by each reanalysis. Detected event dates generally do not vary by more than 1-2 days between reanalyses. The common dates used for this comparison are listed in Table 2.

**Table 2: Dates and types of SSW events used for the comparison. HA and LA denote high-agreement and low-agreement SSW events, respectively.**

| SSW onset date | Event type: split (S) or displacement (D) | Agreement |
| --- | --- | --- |
| 29-Feb-1980 | D | LA |
| 04-Mar-1981 | D | HA |
| 04-Dec-1981 | | HA |
| 24-Feb-1984 | D | |
| 01-Jan-1985 | S | |
| 23-Jan-1987 | D | |
| 08-Dec-1987 | S | LA |
| 14-Mar-1988 | S | HA |
| 21-Feb-1989 | S | |
| 15-Dec-1998 | D | LA |
| 26-Feb-1999 | S | LA |
| 20-Mar-2000 | | |
| 11-Feb-2001 | S | HA |
| 31-Dec-2001 | S | LA |
| 18-Jan-2003 | S | |
| 05-Jan-2004 | D | HA |
| 21-Jan-2006 | D | LA |
| 24-Feb-2007 | D | HA |



| 22-Feb-2008 | D | | LA |
|---|---|---|---|
| 24-Jan-2009 | S | | |
| 09-Feb-2010 | | | |
| 24-Mar-2010 | D | | HA |
| SSW (22) | D (10) S(9) | | LA (7) HA (7) |

SSWs are also known to present a large diversity in terms of how the stratospheric polar vortex is distorted in the course of the events. While some events occur due to a displacement of the vortex, others occur from a splitting (Charlton and Polvani, 2007). These two types of SSW events result from different planetary-scale wave forcing in the stratosphere and can affect the tropospheric flow in different ways (Bancalá et al., 2012; Lehtonen and Karpechko, 2016; Martineau and Son, 2015; Mitchell et al., 2013; Smith and Kushner, 2012). It is thus possible that one type or the other is subject to larger uncertainties in reanalysis datasets. To test whether this is the case, the two types of SSWs, i.e., split SSWs (SSWS) and displacement SSWs (SSWD), are classified using vortex moment diagnostics described in Seviour et al. (2013) (see Table 2).

SSWs are further classified based on whether the reanalysis datasets included in the LRE agree or disagree for a specific diagnostic. Since the Coriolis torque is the forcing term that shows the largest discrepancies among reanalysis datasets in the stratosphere (Martineau et al., 2016), it is used here to quantify the level of agreement. The standard deviation of the Coriolis torque averaged from 45°N to 85°N among the LRE is considered each day from 10 days before the central date to 5 days after the central date which is shown later to be the period when the Coriolis force is largest during SSW events. The agreement is then defined for each event as the maximum standard deviation observed during the period considered. Events are finally classified into two categories: high-agreement SSWs (HASSWs) or low-agreement SSWs (LASSWs) whether they are in the lower 33.3% or upper 33.3% of the agreement index of all SSWs. The type of each event is indicated in Table 2.

## 3 Vortex geometry during the 2009 SSW

Since the geometry of the stratospheric vortex may vary widely from one event to another, the structure of any particular event can be heavily obscured when performing composite means. We therefore first proceed to illustrate and compare the morphology of the stratospheric vortex among reanalysis datasets for a representative SSW event that is well documented in the literature. We choose the January 2009 SSW, an event characterized by a vortex splitting that resulted from an unusually large amount of upward EP flux by wavenumber two (Harada et al., 2010; Manney et al., 2009). Reanalysis datasets show



qualitatively similar downward coupling during this event (Martineau and Son, 2010). Figure 2 illustrates the evolution of geopotential height contours at 10 and 3 hPa. In the course of this event, the polar vortex is progressively elongated to finally split into two individual vortices over northeastern Canada and Russia. The vortex structure features a westward tilt before the onset date (compare geopotential height contours at 10 and 3 hPa), which is consistent with the upward EP fluxes seen

5 during the event (Harada et al., 2010).

The structure of the vortex is generally quite similar among reanalyses at 10 hPa, although some small discrepancies are observed in NCEP-NCAR and NCEP-DOE. Larger differences are found at 3 hPa. Notably, NCEP-CFSR exhibits contours enclosing a smaller area from two days before the onset ($t_{-2}$) and onward. On the other hand, JRA-25 exhibits contours

10 enclosing a larger area, especially from $t_{-2}$ to $t_2$. From inspecting the 2009 SSW events, it is clear that uncertainties in vortex geometry are larger in the upper stratosphere. These differences in geopotential height field are likely accompanied with differences in circulation, and thus, with differences in eddy activity and fluxes among the reanalyses. Although it is not an easy task to summarize differences in vortex geometry for all SSW events, it is possible to evaluate them indirectly through eddy fluxes, which are examined in the following section.





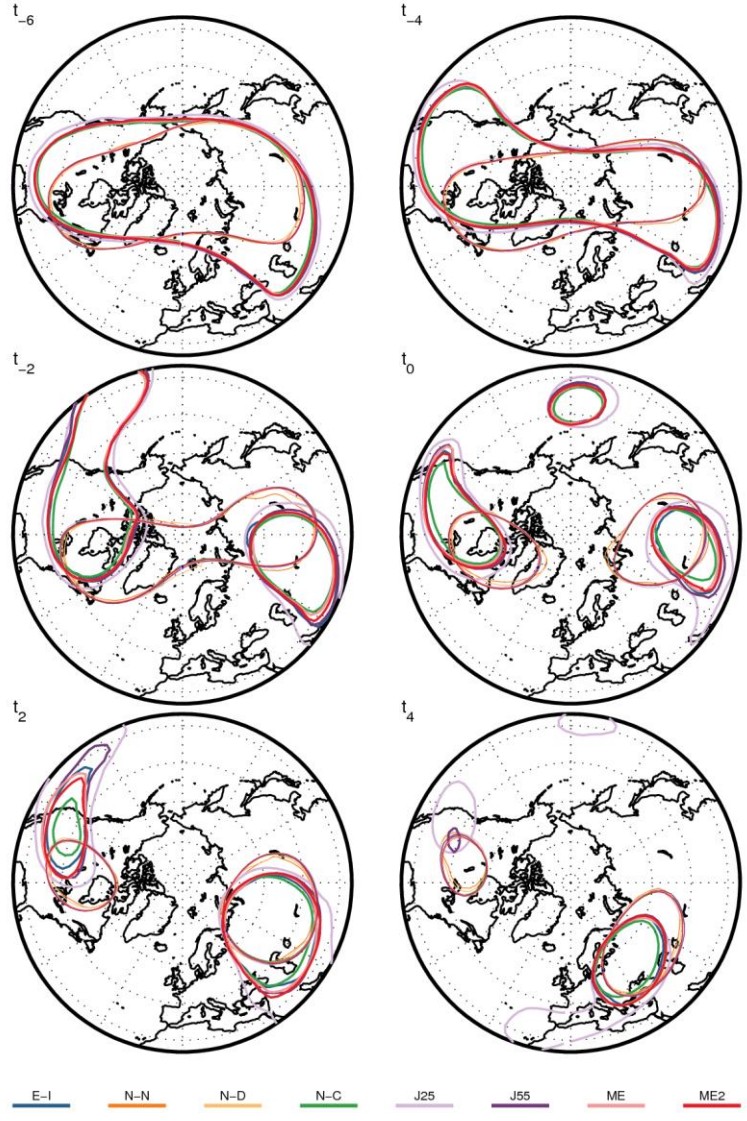

**Figure 2: Vortex geometry at 10hPa (thin) and 3 hPa (thick) during the 2009 SSW for different reanalysis datasets (colours). The 30km contour is illustrated at 10 hPa and the 38.5km contour is shown at 3 hPa. The central date (T$_0$) is set as January 24$^{th}$ 2009.**



## 4 Evolution of zonal-mean flow and eddy fluxes

A composite analysis of the evolution of SSW events is first performed in Fig. 3 by showing the mean and standard deviation of various quantities averaged between 45 and 85°N. Only the LRE members are considered for this analysis. The

evolution of zonal-mean wind highlights the rapid deceleration of the stratospheric polar vortex characterizing SSW events. At the same time, a warming of the stratosphere is observed. The agreement between reanalyses is generally good for both wind and temperature in the lower atmosphere. However, reanalyses show a large spread in $\bar{u}$ (~1m/s) in the upper stratosphere, especially when winds are at their weakest. The spread in $\bar{T}$ also increases towards the upper stratosphere and tends to be larger (~5K) after the occurrence of SSW events. The zonal-mean meridional circulation ($\bar{v}$) becomes

increasingly southward in the high stratosphere before the events, peaking a few days before the reversal of zonal-mean zonal wind. The Coriolis torque resulting from this circulation explains the deceleration of the vortex in the Eulerian framework (Matsuno, 1971). This is in contrast to the transformed Eulerian mean (TEM) framework where the residual circulation is poleward and downward during SSW events (Song and Chun, 2016). Whereas the agreement is good among reanalyses in the troposphere, larger spread is again observed in the upper stratosphere, coinciding with the minimum of $\bar{v}$.

The remainder of Fig. 3 illustrates the evolution of zonal-mean eddy heat ($\overline{v'T'}$) and momentum ($\overline{u'v'}$) fluxes, which are indicative of Rossby wave propagation. Here, both heat fluxes and momentum fluxes are clearly enhanced in the upper stratosphere before the central date, which corresponds to upward and equatorward wave propagation. Again, there is good agreement between reanalyses in the troposphere and lower stratosphere, whereas it is severely degraded in the upper stratosphere. The largest spread occurs coincidentally with the peaks in eddy fluxes and thus with the period when the most

intense Rossby wave propagation occurs.

 Composites of SSW events are not only subject to uncertainties related to the choice of dataset as shown in Fig. 3, but also to uncertainties related to the large diversity of events included in composites. Figure 4 first shows the standard deviation among SSW events for the same quantities shown in Fig. 3. Uncertainties related to the composite methodology are

increasing in the upper stratosphere for all quantities., and they also increase before the onset of SSW events (lag 0), which is the period when these terms have large magnitudes (compare with left column of Fig. 3). Zonal wind shows the largest uncertainty around lag 0 with a minimum at lag 0. This minimum results from the fact that SSW events were defined as a reversal of zonal-mean zonal wind at lag 0, which by construction forces all events to have similar zonal-mean zonal winds at 10 hPa.







**Figure 3: Evolution of zonal-mean variables and eddy fluxes during SSW events (rows) in function of pressure and time. All quantities are averaged from 45ºN to 85ºN. The LRE mean is shown to the left and the LRE standard deviation is shown to the right. Zonal wind ($\overline{u}$) and meridional wind ($\overline{v}$) have units of m/s. Temperature ($\overline{T}$) has unit of K. Heat flux ($\overline{v'T'}$) has unit of mK/s. Momentum flux ($\overline{u'v'}$) has units of m²/s².**





The ratio between the standard deviation among SSW events (Fig. 4 left column) and the standard deviation among datasets (Fig. 3 left column) is then shown in the right column of Fig. 4. This ratio is typically small in the upper troposphere and lower stratosphere, indicating that uncertainty is dominated by the large diversity of SSW events. Except zonal-mean zonal wind, most quantities show enhanced ratios in the upper stratosphere with values of about 0.15. Near the surface, many quantities also show large ratios. This suggests that the uncertainties related to the inter-reanalysis spread have more importance to our interpretation of the evolution of SSW events at the lower boundary and in the upper stratosphere.

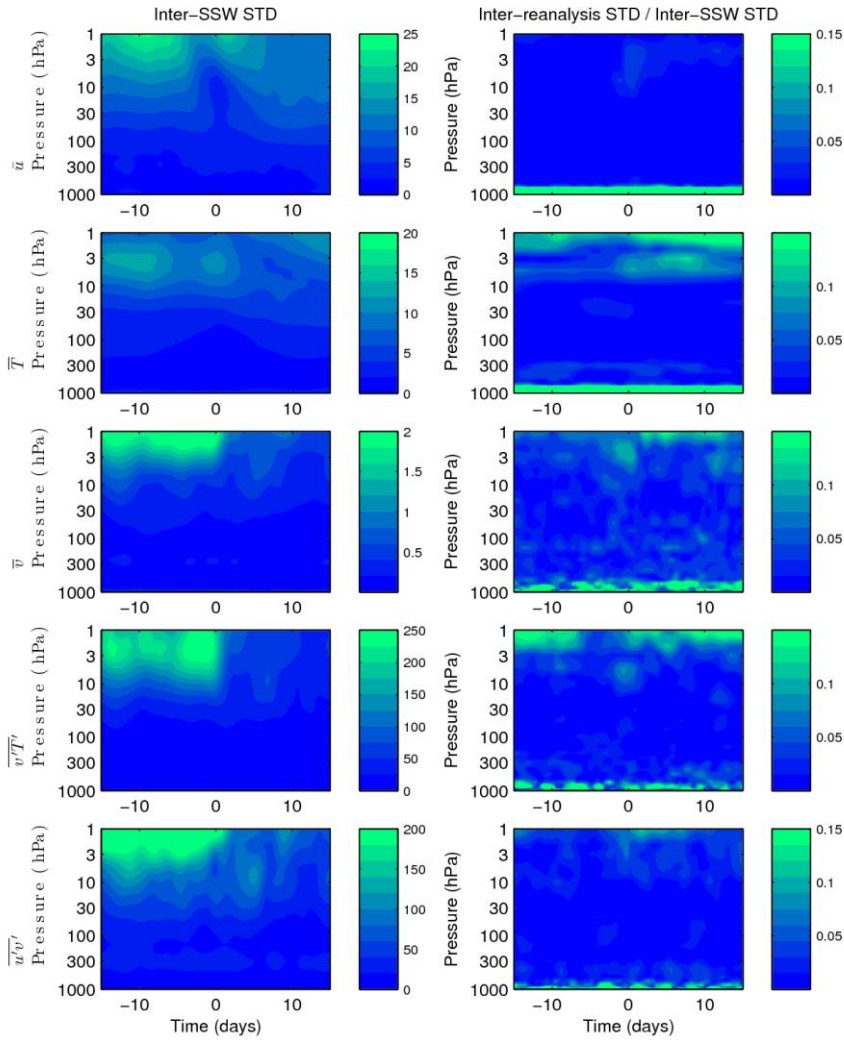

**Figure 4: Similar to Fig. 3 except showing (left) the standard deviation among SSW events and (right) the standard deviation among reanalysis datasets divided by the standard deviation among SSW events.**





The complete budget of zonal-mean momentum is then investigated in Fig. 5. The evolution of each term of Eq. (1.1) is illustrated at two representative levels in the stratosphere: 3 and 10 hPa. The corresponding spread among reanalyses is shown using a logarithmic scale to allow the comparison of uncertainties over a wide range. Most SSW events are characterized by an intense deceleration of the zonal-mean zonal wind for a few days until the central date and immediately followed by a subsequent acceleration. Note the latter acceleration is weaker than the maximum deceleration during the onset, not reverting the zonal wind back to its original strength. The deceleration is most intense at 3 hPa but is also well observed in the mid-stratosphere at 10 hPa. Upper-tropospheric zonal-wind tendencies (not shown) are very weak in comparison to the stratosphere, and therefore almost undetectable if plotted on the same scale. The inter-reanalysis STD increases towards the central date where it peaks at 0.3 m/s/day at 3 hPa. The spread at 10 hPa is typically smaller and peaks around the onset date.

Moving on to the forcing terms of the momentum equation, the Coriolis torque ( $f\overline{v}$ ), a term included in the QG scaling of the momentum equation, shows the largest forcing of all. This term is responsible for a large deceleration of the stratospheric vortex in the upper stratosphere. It peaks at about 2 to 3 days before the onset date and is markedly more intense in the upper stratosphere. Of all the forcing terms, it also shows the largest spread among reanalyses, peaking slightly above 1 m/s/day at 1 to 2 days before the onset date. The uncertainty then decreases at the same time as the forcing itself becomes weaker in the upper stratosphere. In comparison to the upper stratosphere, the inter-reanalysis STD in the mid-stratosphere is smaller and more constant over time.





**Figure 5: (left) Evolution of forcing terms of the zonal-mean momentum equation at 10 hPa (dashed lines) and 3 hPa (solid lines) in the course of SSW events. All variables are averaged from 45ºN to 85ºN. Note that the range of y-axis in each panel is different. (right) The inter-reanalysis spread (standard deviation) of the corresponding terms are shown for the LRE members. The standard deviation is shown on a logarithmic scale: the spacing between tick marks represents a decrease or increase of the standard deviation by a factor of about 3. All quantities are expressed in m/s/day.**





The momentum flux convergence, another term included in the QG scaling of the momentum equation, also shows large forcing in the upper stratosphere. It is largely opposed, but not completely, to the Coriolis torque. Similar to the Coriolis torque, inter-reanalysis STD peaks several days before the onset date (between 0.3 and 1 m/s/day) and is reduced afterwards, especially in the upper stratosphere.

Terms that are left out of the QG form of the momentum equation (Fig. 5, rows 4 to 6) provide much smaller forcing for zonal wind tendency during SSW events in comparison to the convergence of horizontal momentum fluxes and the Coriolis torque. Their differences from one reanalysis to the other are also generally much smaller than those of QG terms. These terms are therefore not a large source of uncertainty in the momentum budget. It is nonetheless worth noting that the convergence of vertical fluxes of momentum is not negligible near the onset date of the event. Forcing magnitudes can reach up to about -2m/s/day. Its inter-reanalysis STD can also be relatively large (up to 0.3 m/s/day) in the upper stratosphere, but is still small in comparison to the other dominant terms of the momentum equation.

The residual, which quantifies the consistency of the momentum diagnostics, is typically negative before the onset. This likely reflects the exclusion of gravity wave drag from the momentum budget (Martineau et al., 2016). Interestingly, its magnitude decreases in the upper stratosphere after SSW events (especially clear in JRA-25). This could be explained by the relatively quiet period following SSW events, when incoming fluxes of planetary-scale waves and gravity waves are supressed (Hitchcock and Shepherd, 2013). Similarly, the residual in the upper stratosphere exhibits a larger spread among reanalyses before the onset date in comparison to after.

The vertical dependence of the forcing terms and their inter-data spread are further investigated using vertical profiles averaged over five days before the onset date (Fig. 6). This period encompasses the period of large inter-reanalysis STD seen in the Coriolis force and momentum flux convergence (Fig. 5). All terms of the momentum equation, wind tendency, and forcing terms show increasingly large magnitudes in the upper stratosphere. Similarly, the inter-data STD is typically small in the troposphere but increases sharply in the stratosphere. Consistent with Fig. 5, terms with the largest inter-reanalysis STD include the Coriolis torque and the momentum flux convergence. Both show a noticeably improved agreement in the stratosphere by considering only the latest reanalysis ensemble (LRE) members instead of all reanalyses. We note that the inter-reanalysis STD becomes more similar between LRE and all datasets above 10 hPa where NCEP-NCAR and NCEP-DOE are left out in the all reanalysis ensemble due to the unavailability of data. This suggests that a substantial fraction of the inter-data STD below 10 hPa is attributable to these two reanalyses.

Compared to QG terms, non-QG terms are smaller in magnitude and agree better. Interestingly, the vertical convergence of momentum fluxes presents a sharp dipole in the vertical near the tropopause. This term, which involves a vertical derivative,



may not be adequately resolved when computed with a coarse vertical resolution. In fact, Martineau et al. (2016) have shown that using more vertical levels reduces the residual in the upper troposphere and lower stratosphere (see their Fig. A1b). In Fig. 6, the residual, R, becomes increasingly negative in the upper stratosphere. Again, this is largely a consequence of the exclusion of parameterized gravity wave drag from the momentum budget (Martineau et al., 2016).

**Figure 6: Vertical profiles of each term in the momentum equation averaged from lags -5 to 0 days during SSW events. All variables are averaged between 45ºN and 85ºN. Individual reanalyses are shown to the left and the inter-reanalysis standard deviation is shown to the right on a logarithmic scale. The latter is shown for all reanalyses (grey) and the LRE members (black). All quantities are expressed in units of m/s/day.**



Each term of the momentum equation and the spread between reanalysis datasets are further evaluated as a function of latitude and pressure in Fig. 7 for the same period shown in Fig. 6 (day -5 to 0). The zonal wind deceleration during this period is characterized by a strong deceleration maximized at about 70°N. A weak acceleration is also present around 30°N in the upper stratosphere, resulting primarily from the Coriolis torque (Martineau and Son, 2015). Zonal wind tendency is fainter in the troposphere. As mentioned earlier, the agreement between datasets is much better in the lower atmosphere than the middle atmosphere. Large discrepancies are limited to the upper stratosphere and peak where the deceleration of the vortex is strongest.

The QG terms, as expected, are strongly opposed to each other in both the troposphere and the stratosphere. The Coriolis torque is responsible for most of the deceleration of the stratospheric vortex. As mentioned above, it is also responsible for zonal wind acceleration in the midlatitudes and subtropics. The efficiency of this forcing depends on the extent to which it is opposed by the convergence of fluxes of momentum. While they are strongly opposed in the upper troposphere, they are not perfectly balanced in the stratosphere, which results in the observed zonal wind tendencies. While the deceleration of the stratospheric vortex and its forcing is observed poleward of 50°N and thus well captured by averaging poleward of 45°N, QG forcing terms show a strong tripole in the troposphere, associated with the Hadley, Ferrel and polar cells, which results in some unavoidable cancellation of the forcing when averaging from 45 to 85°N, explaining why we do not observe large QG terms in the troposphere in Fig 6. The spread in the QG terms is maximized in the high-latitude stratosphere and a substantial reduction of the spread between datasets is evident in the lower stratosphere.

As expected, non-QG terms show much smaller forcing and inter-data spread in the high latitudes. Interestingly, advection terms are maximized in the subtropical upper stratosphere and tropopause. Among the non-QG terms, only the vertical convergence of momentum fluxes shows substantial forcing for deceleration in the upper stratosphere in the mid-latitudes. It also displays large and sharp forcing at the tropopause, a feature observed clearly in averages over the high-latitudes (Fig. 6). The resulting residual is typically negative and is also maximized in the high latitudes. On the other hand, it is larger in the jet's vicinity in the troposphere. The largest inter-data STD of the residual is in the polar region in the upper stratosphere.





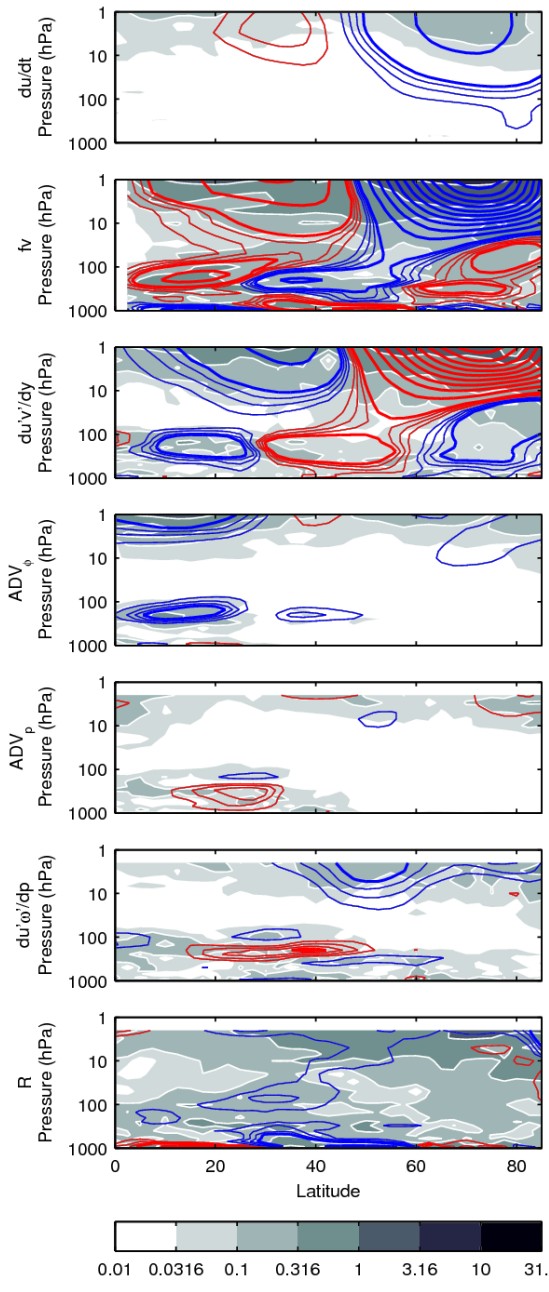

**Figure 7: Pressure-latitude cross-section of inter-reanalysis mean terms of the momentum equation averaged from lags -5 to 0 days during SSW events. Composite mean is shown with red or blue for positive or negative values, with a contour interval of 0.25 for values ranging from 0.5 to 1.5 with thin contours and intervals of 2.5 ranging from 2.5 and larger with thick contours. The inter-reanalysis standard deviation is shaded per the colour bar. All quantities are shown for the LRE members and are expressed in units of m/s/day.**



The inter-dataset spread is further highlighted for QG terms in Figs. 8 and 9. The two figures show the difference between each reanalysis and the mean of the LRE members. The large-scale structure of the Coriolis torque (Fig. 8) is generally similar among reanalyses but notable differences are observed in the upper stratosphere. Particularly, both ERA-Interim and NCEP-CFSR show weaker forcing for deceleration (positive bias) in the high-latitude upper stratosphere in comparison to

5   others. Looking at the same cross-section for the momentum flux convergence (Fig. 9) it is noticeable that there is generally a better agreement with the mean of the LRE members in the lower stratosphere and troposphere. However, differences are still large in the upper stratosphere. Interestingly, the spatial distribution of biases in the Coriolis torque and the momentum flux convergence are somewhat opposed in the upper stratosphere in some reanalyses which may be indicative of a compensation between biases of the two leading terms of the momentum equation.

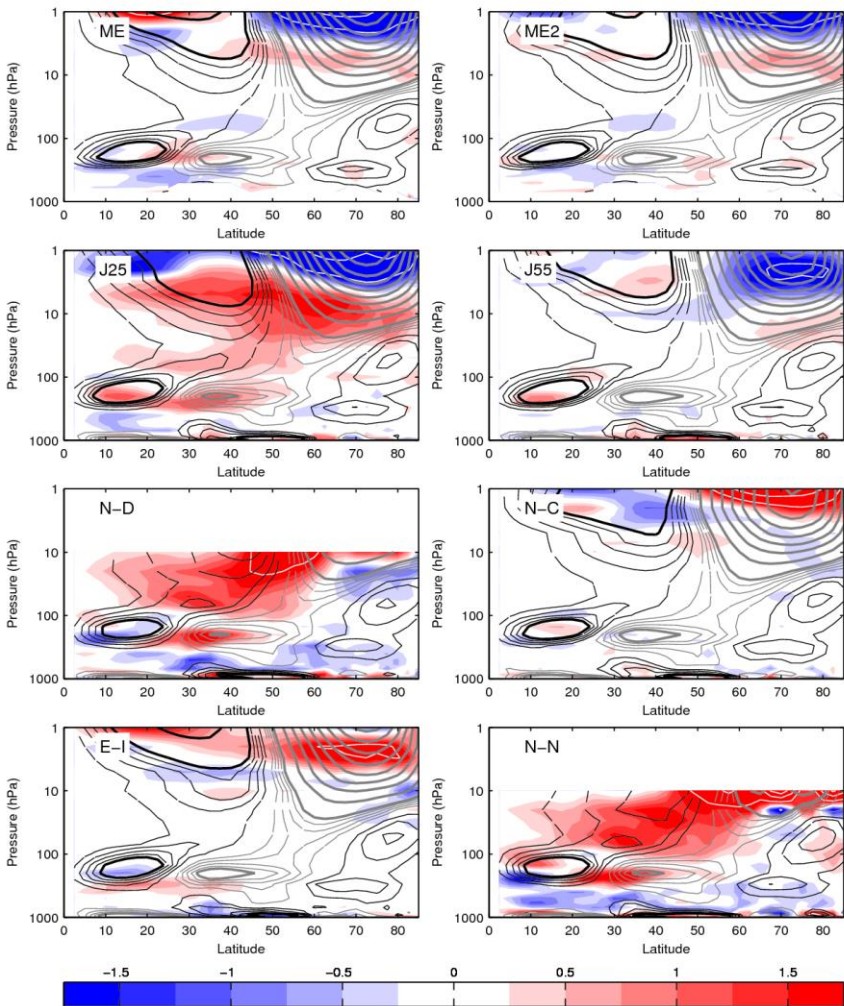

**Figure 8: Latitude/pressure cross-section of the Coriolis torque averaged from lag -5 to 0 days during SSW events for different reanalysis datasets. The positive and negative values are contoured in black and grey, respectively. Thin contours range from -4 to**



**4 m/s/day in steps of 1 m/s/day and thick contours for 5 m/s/day and larger in steps of 5 m/s/day. Biases with respect to the mean of LRE members are illustrated with red and blue shading for positive and negative values, respectively. The colour interval is 0.25 m/s/day from -1.75 to 1.75. Values larger (smaller) than 2 (-2) are contoured every 2 m/s/day in white.**

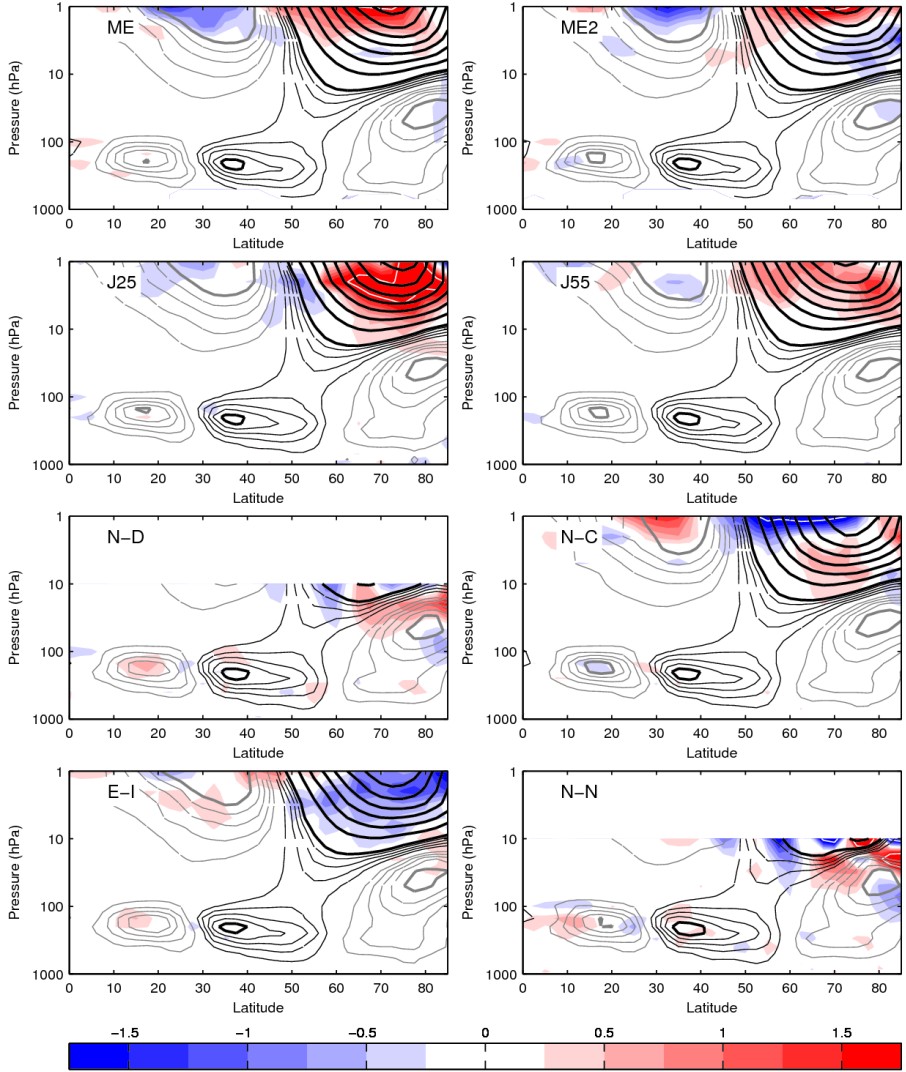

**Figure 9: Same as Fig. 8 but for momentum flux convergence.**

In order to identify possible sources of inconsistency in the momentum equation, the relationship between the residual and the dominant forcing terms is explored in Fig. 10 at three representative levels (3, 10 and 300 hPa). This analysis relies on the assumption that the inter-reanalysis variability of R is due to differences in the forcing terms and not to differences of zonal-wind tendency. This assumption is approximately valid since discrepancies in zonal wind tendencies are generally much smaller than discrepancies in the dominant terms of the momentum equation (see Figs. 5, 6 and 7). Similar to





Martineau et al. (2016), a large fraction of the variability in the residual among reanalyses in the mid-stratosphere (10 hPa) can be attributed to the Coriolis torque (r=-0.99). Some relationship, although less significant, is also seen between the residual and the momentum flux convergence at 10 hPa (r=-0.80). Added together, the QG terms can explain all of the variability of the residual (r=-1). While this high correlation owes partly to the fact that JRA-25 is an outlier, NCEP-NCAR

5   and NCEP-DOE also hint to a strong relationship between QG terms and the residual by inspecting the QG residual ($R_{QG}$), which is computed by excluding non-QG terms from Eq. (1.1). The dominant role of the Coriolis torque is not as evident in the upper stratosphere (3 hPa) and the upper troposphere (300 hPa) but still plays an important role (r=-0.57 and r=-0.85, respectively).

10   The scatterplots highlight a convergence of newer reanalyses datasets in some circumstances. This is mostly apparent at 10 hPa where ERA-Interim, NCEP-CFSR, JRA-55, MERRA and MERRA2 are strongly clustered together. At 300 hPa, however, MERRA and MERRA2 tend to be apart from other reanalyses and at 3 hPa, JRA-55 and ERA-Interim tend to differ from the others. One should therefore not assume that there is always convergence when considering the latest datasets.

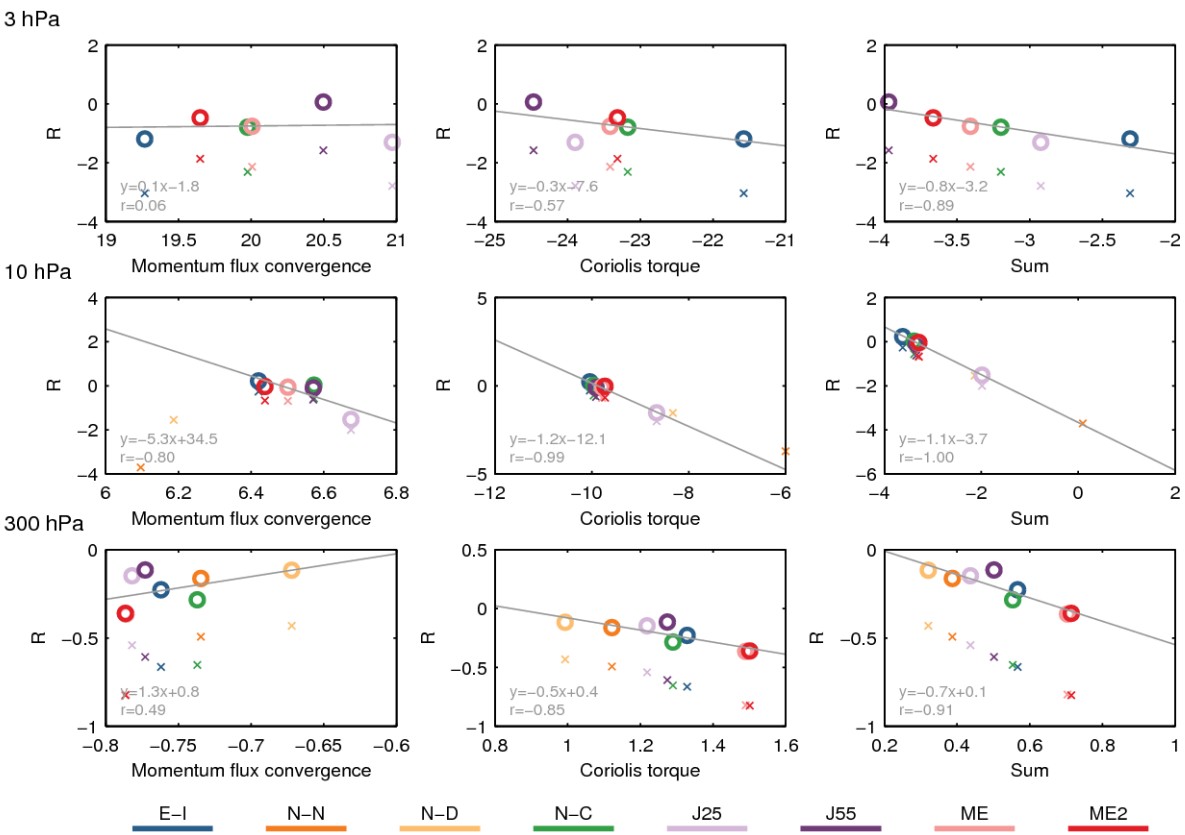





**Figure 10: Scatterplot comparing the residual (R) with respect to (left) momentum flux convergence, (centre) Coriolis torque, and (right) their sum. While R is shown with circles, $R_{QG}$ is shown using crosses. All variables are averaged between 45ºN and 85ºN and from lag -5 to 0 during SSW events. The (top) 3 hPa, (middle) 10 hPa, and (bottom) 300 hPa levels are shown. All variables are expressed in units of m/s/day. Linear regression of R with respect to each forcing term is displayed in each panel.**

To further explore a cancellation between the dominant terms of the momentum equation, the linear relationship between the Coriolis torque and the momentum flux convergence among reanalyses is also evaluated at the same three representative pressure levels (300, 10 and 3 hPa, not shown). While they are weakly opposed at 10 hPa and 300 hPa, there is a clear and strong anti-correlation between the two forcing terms at 3 hPa (r=-0.82). Reanalyses that show an enhanced deceleration by the Coriolis torque typically exhibit an enhanced forcing for acceleration by momentum flux divergence. Although the compensation is not perfect (slope of -1.3 when regressing Coriolis torque on momentum flux convergence), this compensation helps nonetheless to reduce the residual of the momentum equation, but does so at the expense of the accuracy of the forcing terms of the momentum equation.

To further identify causes of inter-reanalysis discrepancies, we now proceed to compare SSW events with high agreement between reanalyses (HASSWs) and those with low agreement (LASSWs). The classification of these events was described in Section 2.3. Figure 11 shows several momentum budget terms at 10 hPa during both event types. It is found that LASSWs are markedly more intense than HASSWs as seen by the peak deceleration of zonal wind that is twice as strong around the onset of the SSWs. The forcing terms are also markedly larger during LASSWs. Despite the larger forcing terms, a strong cancellation is observed between the Coriolis torque and momentum flux convergence. On the other hand, HASSWs show only steady and moderate forcing. To investigate the role played by wave drag in both types of event, we illustrate EP flux convergence and the contribution of zonal wavenumber-1 planetary waves (see Appendix B of Martineau et al., 2016 for a formulation of EP flux and TEM momentum equation in pressure coordinates). We find that LASSWs present substantially larger convergence of wave activity fluxes in the stratosphere in comparison to HASSWs. Most of the difference in wave drag can be attributed to wavenumber-1 EP flux divergence.

Among the reanalyses, JRA-25 seems to stand out from the others in terms of EP flux divergence by wave-1 during LASSWs. Wave drag is especially affected by a positive bias at the early stage of these events. This difference could be due to a bias in stratospheric temperatures affecting the computation of heat fluxes and static stability, terms included in the computation of the vertical component of EP-flux. This bias is subsequently corrected in JRA-55 (Kobayashi et al., 2015), which could explain the better agreement between JRA-55 and other members of the LRE. Another visible outlier, ERA-





Interim, underestimates wave drag during LASSW events. These discrepancies are observed in many events included in the composite (not shown).

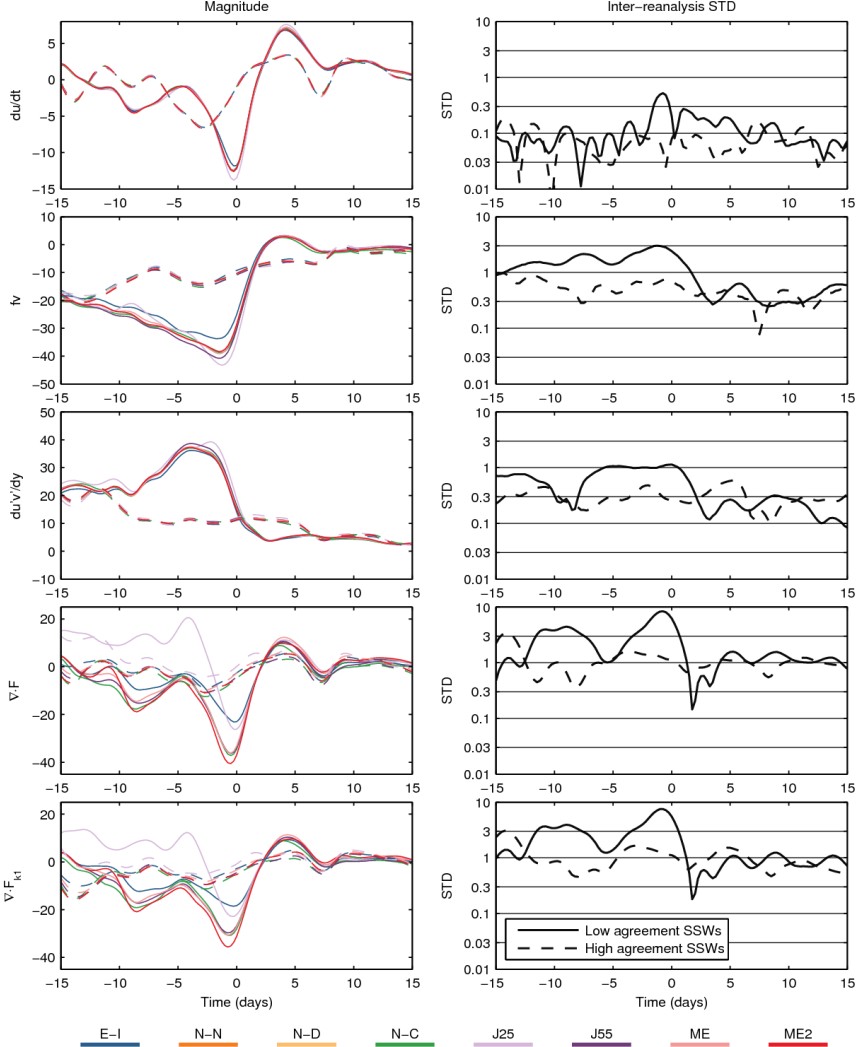

**Figure 11: Similar to Fig. 5 but for comparing SSWs with small (HASSWs – dashed lines) and large uncertainties (LASSWs – solid lines). EP flux divergence is shown for all wave numbers (4th row) as well as wavenumber-1 (bottom row). All diagnostics are shown at 10 hPa and are in units of m/s/day.**

Next, the difference in the propagation of planetary-scale waves between HASSWs and LASSWs is illustrated in Fig. 12. LASSWs present markedly stronger forcing for deceleration (EP flux convergence) in the upper stratosphere in comparison to HASSW events and overall stronger inter-reanalysis uncertainty. Although LASSW events show both stronger upward and equatorward EP flux, most of the uncertainty is related to the vertical propagation alone. It is possible that these



differences in wave drag among reanalyses be a source of large discrepancies in Coriolis torque through the generation of a compensating residual circulation (Song and Chun, 2016). However, as discussed earlier, the inter-reanalysis variability in momentum fluxes does not fully account for the variability in Coriolis torque (Fig. 11) in LASSWs. This suggests that other factors, such as gravity wave drag and radiative transfer, could also play a role. Despite the larger discrepancies in

5     stratospheric circulation arising during LASSW events, there is no evidence of increased uncertainty of the tropospheric circulation in these events (not shown), indicating that either these differences have negligible impact on stratosphere-troposphere coupling or that any differences in downward coupling are prevented by data assimilation in the troposphere.

Since the previous analysis hinted that wave-1 fluxes of wave activity are an important source of uncertainty among

10     reanalyses, we also compared displacement (SSWD) and split (SSWS) events to evaluate whether vortex geometry could be a source of uncertainty (not shown). The two types of SSWs show quite similar uncertainties among the various forcing terms. In fact, there is no clear separation between SSWD and SSWS; although SSWD events show more intense EP flux divergence by wavenumber-1, both types are forced by wavenumber-1 (Bancalá et al., 2012). Wavenumber-2 could simply show less uncertainty because it plays a lesser role in the composites shown. Among a total of seven LASSWs, four events

15     are classified as SSWD events whereas three events are SSWS events and out of seven HASSWs four and two events are SSWDs and SSWSs, respectively. The remaining events are not clearly classified in either category. These statistics suffer however from the small sample sizes and there is no clear preference for split or displacement SSW events to be better represented in reanalysis datasets, as far as the inter-dataset agreement is concerned.





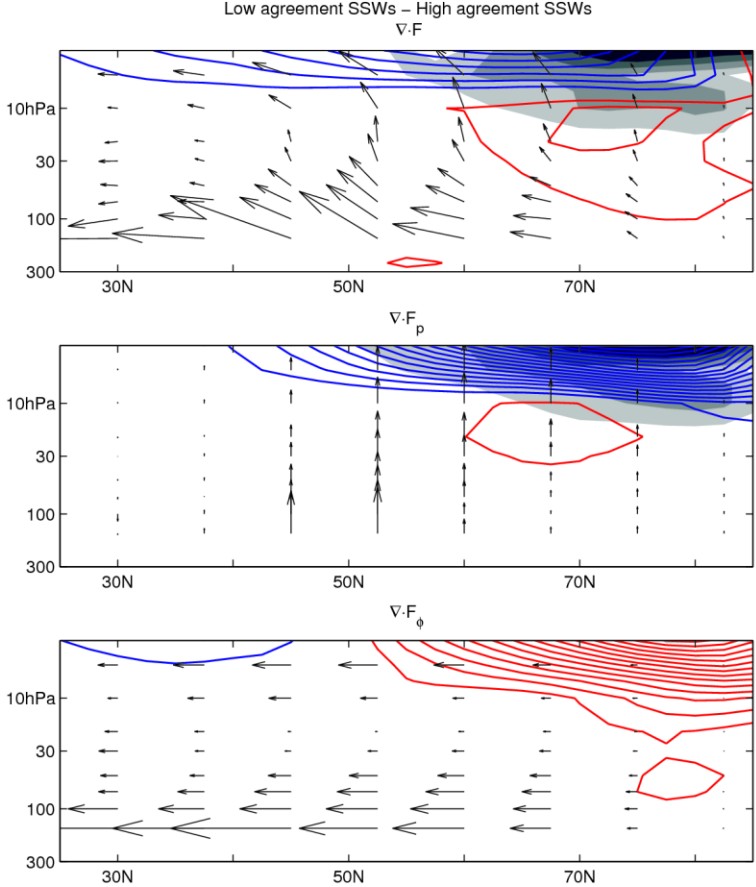

**Figure 12: Difference in EP flux (arrows) and its divergence (contours) between HASSWs and LASSWs. All quantities are averaged over lags -7 to 2 days. EP flux convergence and divergence, respectively, are contoured in red and blue with an interval of 2.5 m/s/day. Differences in inter-reanalysis spread are shown in grey with a shading interval of 1m/s/day. Only LRE members are considered in this figure.**

## 5 Summary and conclusions

To assess uncertainties in the dynamical variability of the stratosphere and troposphere and their coupling in reanalysis datasets, a detailed comparison of zonal-mean momentum diagnostics is carried out for eight reanalysis datasets during sudden stratospheric warming events (SSWs). Emphasis is placed on the vertical and temporal dependence of the uncertainties during the events as well as on the factors that lead to the uncertainties.

From the troposphere to the mid-stratosphere, all quantities of the momentum equation are remarkably similar among the datasets. Although inter-data discrepancies increase substantially towards the upper stratosphere, zonal-mean zonal wind and temperature, often used to illustrate the vertical coupling during SSW events, agree quite well up to the mid-stratosphere. As




such, the temporal-spatial evolution of composite SSW events is nearly identical in the different datasets (Martineau and Son, 2010; Palmeiro et al., 2015).

Non-negligible uncertainties are observed mainly in the upper stratosphere. They are particularly large during the most

intense SSW events, indicating that uncertainties in the momentum tendency and the related eddy fluxes are related to the strength of the episodes of planetary-scale wave propagation from the troposphere to the stratosphere. No significant difference among reanalyses, however, is found when comparing splitting and displacement events.

Among all forcing terms of the zonal-mean momentum equation, the Coriolis torque and the meridional convergence of

momentum fluxes show the largest magnitudes and largest disagreement among reanalyses several days before the reversal of zonal wind in the stratosphere. Such uncertainties decrease dramatically after the zonal-mean winds change direction during SSW events. This could be explained by the fact that a dynamically quiet period generally follows SSW events (Hitchcock and Shepherd, 2013). Other forcing terms, i.e., non-QG terms, show smaller magnitude, and smaller inter-dataset uncertainty in comparison to the Coriolis torque and the convergence of momentum fluxes.

The large variability of forcing terms in the high stratosphere among reanalyses exceeds many times the uncertainties in zonal wind. Thus, some reanalysis datasets exhibit large residuals in the momentum equation. Interestingly, the residuals are large and vary substantially among datasets prior to the reversal of zonal wind. This is consistent with the enhanced residual observed in periods of vortex transience in the analysis of Martineau et al. (2016). A marked reduction of the magnitude of

the residual after the reversal of the zonal-mean circulation again suggests a less dynamically active period where the residual may be decreased in part because of a reduction of gravity wave drag in the upper stratosphere (Hitchcock and Shepherd, 2013). Most of the residual in the stratosphere is correlated to uncertainties in the Coriolis torque which may indicate that the residual circulation responds in a way as to balance missing forcing in the momentum equation. The same also holds in the troposphere. Unlike the mid-stratosphere and troposphere, however, the residual of the zonal-mean

momentum equation in the upper stratosphere benefits from a cancellation between biases in the Coriolis torque and eddy momentum flux convergence. Although this phenomenon contributes to a seemingly improved momentum budget, it does not help to reduce the uncertainties in the dynamical evolution of the events. This uncertainty does not, however, overly alter our interpretation of the dynamics regulating SSW events and should therefore not be a concern when studying SSW events in the troposphere and stratosphere. This relationship also indicates that a fraction of the variability of the meridional

circulation among reanalyses is driven by inter-dataset differences of eddy fluxes.

A reduction of the inter-reanalysis spread during SSW events is generally observed in newer reanalyses for most terms of the momentum equation, especially in the stratosphere. Inspection of individual reanalyses however reveals that newer reanalyses are not always clustered together and that outliers vary depending on the pressure level and the variables.



Since this analysis compares the momentum diagnostics among reanalysis datasets, the uncertainties discussed here are resulting from both the evolution of the different atmospheric fields by the forecast process and their subsequent adjustment by the data assimilation process. Discrepancies among reanalyses may thus originate from differences in the models, observations being assimilated, and assimilation techniques. Sources of error also include processes that are not well captured or parameterized, the latter not being considered in the momentum budget of this study. As discussed in Martineau et al. (2016), some fields may be easier to constrain than others, and thus more representative of the true evolution of the atmosphere. For instance, the zonal-mean zonal wind may be better constrained by temperature observations by assumptions of balance, like thermal wind balance. On the other hand, ageostrophic flows such as the meridional circulation, may be harder to constrain by data assimilation. It is possible that this loosely-constrained circulation may act to oppose biases in other forcings. A better understanding of the distinct contributions of the modelling and data assimilations steps to the observed uncertainties would require studying the forecasts and analysis increments separately.

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
