# Peer review of "A comparison of the momentum budget in reanalysis datasets during sudden stratospheric warming events"

_Atmospheric Chemistry and Physics, 2017_

## Referee Comment (RC1) · Anonymous Referee #2 · 29 Nov 2017

**General comments**

This is an interesting study comparing several reanalysis datasets to analyze the momentum equation during sudden stratospheric warmings (SSW). In particular, I find useful the idea of analyzing the latest reanalysis ensemble separately. Having so many available datasets, an evaluation of how they perform depending on the topic of the study is required. I have enjoyed reading the manuscript and I appreciate the advises and recommendations regarding the uses of different reanalysis in stratospheric dynamics. The contain of this manuscript would be valuable for the scientific community so I recommend it for publication. I only have some minor questions listed next.

**Specific comments**

Page 6, line 10: What happens if a SSW is detected in less than 4 datasets? Is it excluded? Does this happen in many cases?

Table 2: Why the analysis stops in 2010? There is no data for some reanalysis considered herein?

Page 7, line 5 and Page 24, last paragraph: The authors argue that split and displacement SSW-types result from different planetary-scale wave forcing which motivates that distinction. I wonder whether it would be more meaningful to distinguish between wavenumber 1 and 2 as SSW precursors instead of the split/displacement sub-classification. As shown in previous studies (e.g., Bancala et al. JGR-2012), the ratio W2/W1 events will be much smaller than the split/displacement type, but it could be worthy to check whether differences are significant in that case.

Page 12, line 8: What do you mean with "our interpretation of the evolution of SSW events"?

**Technical corrections**

Figure 2: It is difficult to distinguish the thin and thick lines. Maybe using dashed and solid lines as in figure 5?

Figures 3 and 4: I have found difficult to distinguish contours with this color scale. Especially in a printed version.

Figure 6 (right column): Note that some plots are out of range.

Page 8, line 9: events or event?

Page 10, line 25: quantities.,

Page 15, 4: Here, and in other parts of the text: m/s/day. This notation is confusing, I would suggest using $\text{ms}^{-1}\text{day}^{-1}$

---

## Referee Comment (RC2) · Anonymous Referee #3 · 31 Jan 2018

General Comments: This paper examines and compares the momentum budget during sudden stratospheric warming (SSW) events using eight reanalysis data sets. Their results provide some insights into the uncertainties of the budget equation during SSWs, especially the contributions of the QG and non-QG terms, the spread or the discrepancies in terms of the regions and the periods. It is also very useful to know that the spread is much reduced in the latest reanalysis products.

The authors suggested that the largest discrepancy originated mainly from the Coriolis torque (in abstract, line 13, page 7 and section 5). Momentum flux convergence is mentioned as the second term which presents non-negligible spread. I am concerned

the word "originated". This gives an impression that if we fix fv, we would get SSW right. However, the origin of the uncertainties must be in the wave forcing rather in the zonal mean meridional velocity, given the meridional circulation is driven primarily by wave forcing.

Also, what their results actually suggest is that the largest discrepancy is associated with the residual term R, the last term in equation (1.1). This can be seen clearly in their figures 5-7. The standard deviation associated with R is slightly smaller but comparable in magnitude to that of fv. However, the mean state of fv is one magnitude larger than R. Thus, R rather than fv has the largest discrepancies. I suggest that the authors make this point clearer by simply stating that the resolved part of the discrepancies is mainly associated with fv.

I am concerned with their definition of high-agreement and low-agreement SSW events. Those events were defined by the standard deviation of the Coriolis torque averaged from 45-85N. fv is not even an effective measure for SSW events. There are times (within ∼5-15 day average window) when zonal mean fv is large but there is no SSW. I do not think that it is appropriate to define the strength of a SSW event (i.e the strongest or weakest) or the associated discrepancies just by using fv. Again, this is because both SSW and the changes in fv and their associated uncertainties are consequences of wave mean-flow interaction.

Other than the above points, the paper is well written in general. I suggest publication with some effort to improve the clarity of the expressions. More specific comments are provided below.

Specific comments:

1) Line 15, page 1: "the onset of SSW events, a period characterized by unusually large fluxes of planetary-scale waves from the troposphere to the stratosphere". This sentence holds true only if the period is ∼40 days (Polvani and Waugh 2004). The correction between the wave fluxes (or v'T') would become much reduced if the averaging

period is only 5-15 days, which is used in this study (i.e. figures 7-8 and figures 11-12). At these shorter time scales, stratospheric internal variation becomes important. This is precisely why the models cannot predict the timing or the initialization of SSWs. The authors must be careful when they discuss their results and when they related to the EP flux divergence to those from the troposphere.

2) Line 20, page 1: "The strongest SSWs being subject to larger discrepancies among reanalyses". This sentence gives one impression that there is an accepted definition of "the strongest SSWs". Naturally, the readers would think that these events produced the warmest temperature or strongest easterly winds. Is this true?

3) Line 15-16, page 3. It is better to state that the previous assessment was mainly for the extratropics. In the tropics where the QBO becomes important, higher vertical and horizontal resolution should lead to much improved dynamical consistency.

4) Line 21-22, page 4. The last term R also accounts for non-conservative processes, such as Rossby wave breaking (RWB). During SSW, planetary-scale RWB can play an important role. Interestingly, the largest error is associated with R rather than fv term.

5) Lines 12-16, page 7. Now I understand that the definition is based on the largest discrepancies in the Coriolis torque. This needs to be made clearer in the abstract when you mentioned the strongest SSWs because there is no such a definition in terms of the known or accepted description of the SSWs. Also, see my general comments for further concerns.

6) Line 2, page 8. "The evolution of geopotential height contours". Please include the values here (not just in the figure caption) and justify why those values are used to describe the polar vortex. Ertel Potential vorticity should be a much better quantity for this purpose and why not to use EPV?

7) Figures 3-4. It is really hard to qualify the spread or discrepancies based on the color bar used.

8) Line 7, page 15. "Terms that are left of the QG from of the momentum equation provide much smaller forcing for zonal wind tendency during SSW events ... Their differences from one reanalysis ". I disagree for the following reasons. 1). The two QBO terms are of the opposite sign in general (see figure 7). If they are added together, the sum would have a comparable magnitude when it is compared with the other terms. 2). It is well-known that the SSW events often involve breaking of finite amplitude waves. Such an effect cannot be accounted for by 2.5 resolution pressure level data. Please reword the part to avoid the possibility of misleading the readers. See my general comments for further information.

9) I am not sure whether or not figures 8 and 9 is needed. Would it be more concise or informative if the figures were combined as one and show the two groups: the latest versus older generation reanalysis products?

10) Lines 18-28, Page 22. I suggest that the authors to check would the same spread or results be obtained using the residual term R and its standard deviation to define HASSWs and LASSWs. Same applies to figures 11 and 12.

11) Line 14, page 26. See general comments. The results do not suggest that the discrepancies in those non-QG terms are smaller than the QG-terms.

12) Line 23, page 26. "Most of the residual in the stratosphere is correlated to uncertainties in the Coriolis torque". This is very interesting and somehow expected. My explanation is as follows. In the upper stratosphere, gravity wave breaking and finite amplitude wave activities appear regularly there but their propagation cannot be well captured by 2.5 degree pressure level data. Their effects on the polar vortex or zonal mean zonal wind would be included in R or the vertical momentum flux term especially when the QG-terms are calculated by using variables such as u and v, as it is done by this study. On the other hand, when the EP flux divergence is included as in the transformed Eulerian mean equations, the variation of wave forcing would be better resolved by the data used. This is because Del F accounts for the vertically propagating

wave not just the meridionally propagating waves. This is confirmed by figure 11. The figures shows, at 3 hPa, the temporal evolution of the zonal mean wind tendency follows better with the EP flux divergence, less so in terms of fv. Thus, I would think that it is the uncertainties associated with non-resolved wave forcing caused the spread in fv, rather than the other way around.

Minor comments: 1) Line 29, page 1. Too many citations here for motivation. 2) Line 5, page 2. Two daughter vortices -> two vortices. 3) Line 6, page 2. Please be more specific about the differences. Otherwise, delete the sentence as it adds no information. 4) Line 9, page 2. "the general signature". What is it? Please be more specific. 5) Line 25, ., -> , 6) Line 16, high stratosphere -> upper stratosphere.

---

## Author Response (AR1)

**Referee 2**

We are thankful to the reviewer for these comments which help us improve the quality of our manuscript. Our response to each comment is written below. We note that we have noticed a small coding error: averages were made between 50-85N instead of 45-85N as indicated in the manuscript. After correcting this mistake, there is a small change in the classification of High-Agreement and Low-Agreement events (please see the correction to the table where SSW events are listed) but this does not alter our interpretation of the results.

General comments

This is an interesting study comparing several reanalysis datasets to analyze the momentum equation during sudden stratospheric warmings (SSW). In particular, I find useful the idea of analyzing the latest reanalysis ensemble separately. Having so many available datasets, an evaluation of how they perform depending on the topic of the study is required. I have enjoyed reading the manuscript and I appreciate the advises and recommendations regarding the uses of different reanalysis in stratospheric dynamics. The contain of this manuscript would be valuable for the scientific community so I recommend it for publication. I only have some minor questions listed next.

Specific comments

Page 6, line 10: What happens if a SSW is detected in less than 4 datasets? Is it excluded? Does this

happen in many cases?

If the event is not detected in at least 4 datasets it is excluded, which we now indicate in the manuscript. It happens in February 81, February 95 and February 2002. This means only three events out of a total of 25.

Table 2: Why the analysis stops in 2010? There is no data for some reanalysis considered herein?

The last SSW listed is in 2010 because we use the official comparison period of the S-RIP project which ranges from 1980 to 2012 (mentioned in the data section). No events are detected in 2011 or 2012. Only one more event would be included if the analysis is extended to 2013 after which JRA-25 is not provided.

Page 7, line 5 and Page 24, last paragraph: The authors argue that split and displacement SSW-types result from different planetary-scale wave forcing which motivates that distinction. I wonder whether it would be more meaningful to distinguish between wavenumber 1 and 2 as SSW precursors instead of the split/displacement sub-classification. As shown in previous studies (e.g., Bancala et al. JGR-2012), he ratio W2/W1 events will be much smaller than the split/displacement type, but it could be worthy to check whether differences are significant in that case.

Thank you very much for the comment. It is true that the geometry of the vortex (split or displacement) does not necessarily reflect the wave drag that produced the SSW event. To investigate the role of the longitudinal scale of wave activity fluxes to the spread among reanalysis, we classify events according to whether they are dominated by wavenumber-1 (W1) EP flux or wavenumber-2 (W2) EP flux. We indicate the outcome of this classification in Table 2 of the revised manuscript.

We have added the following paragraph in the manuscript: *On the other hand, when considering the dominant fluxes of wave activity producing SSW evens, we find that out of 7 LASSWs, four are W1-dominant and 1 is W2 dominant and out of 7 HASSWs, 1 is W1-dominant and 3 are W2-dominant. This seems to indicate that wavenumber-1 wave drag is responsible for larger uncertainties in reanalysis datasets but a detailed analysis reveals that inter-reanalysis spread is not markedly different between W1-dominant and W2-dominant events (not shown) suggesting that it is the intensity of wave drag rather than the longitudinal scale of wave activity that is linked to uncertainties among reanalyses.* Supplementary Fig. 1 shown here supports this analysis.

[Figure]

**Supplementary Figure 1: Similar to Fig. 11 of the manuscript but for comparing SSWs that are primarily forced by wave-1 or wave-2 EP flux (solid and dashed, respectively; shown on left side) and SSWs that are displacements or splits (solid and dashed, respectively; shown on right side). Although there are small differences in the deceleration of zonal-mean zonal wind, the weakest stratospheric winds following the SSWs are similar in all types of events (not shown).**

Page 12, line 8: What do you mean with "our interpretation of the evolution of SSW events"?

What we mean is that the uncertainties in the upper-stratosphere have a greater impact on how we understand the evolution of SSW events from observations. We rephrase this section to improve the clarity.

Technical corrections

Figure 2: It is difficult to distinguish the thin and thick lines. Maybe using dashed and solid lines as in figure 5?

We are now using dashed lines.

Figures 3 and 4: I have found difficult to distinguish contours with this color scale. Especially in a printed version.

We use a new color scale that makes interpreting these figures easier.

Figure 6 (right column): Note that some plots are out of range.

We extend the range of this figure.

Page 8, line 9: events or event?

Corrected

Page 10, line 25: quantities.,

Corrected

Page 15, 4: Here, and in other parts of the text: m/s/day. This notation is confusing, I would suggest using ms-1day-1

We now use ms$^{-1}$day$^{-1}$ throughout the text.

**Referee 3**

We are thankful to the reviewer for these comments which help us improve the quality of our manuscript. Our response to each comment is written below. We note that we have noticed a small coding error: averages were made between 50-85N instead of 45-85N as indicated in the manuscript. After correcting this mistake, there is a small change in the classification of High-Agreement and Low-Agreement events (please see the correction to the table where SSW events are listed) but this does not alter our interpretation of the results.

General Comments: This paper examines and compares the momentum budget during sudden stratospheric warming (SSW) events using eight reanalysis data sets. Their results provide some insights into the uncertainties of the budget equation during SSWs, especially the contributions of the QG and non-QG terms, the spread or the discrepancies in terms of the regions and the periods. It is also very useful to know that the spread is much reduced in the latest reanalysis products.

The authors suggested that the largest discrepancy originated mainly from the Coriolis torque (in abstract, line 13, page 7 and section 5). Momentum flux convergence is mentioned as the second term which presents non-negligible spread. I am concerned the word "originated". This gives an impression that if we fix fv, we would get SSW right. However, the origin of the uncertainties must be in the wave forcing rather in the zonal mean meridional velocity, given the meridional circulation is driven primarily by wave forcing.

We agree with the reviewer that the choice of the word "originated" is not appropriate. The discrepancies in the Coriolis force may be due to discrepancies in wave drag (resolved and not resolved) but also due to biases in the mean state and data assimilation procedure (Kobayashi and Iwasaki, 2016; Uppala et al., 2005). We rephrase our statement to: While the largest uncertainties in the momentum budget are found in the Coriolis torque, momentum flux convergence also presents a non-negligible spread among the reanalyses.

Also, what their results actually suggest is that the largest discrepancy is associated with the residual term R, the last term in equation (1.1). This can be seen clearly in their figures 5-7. The standard deviation associated with R is slightly smaller but comparable in magnitude to that of fv. However, the mean state of fv is one magnitude larger than R. Thus, R rather than fv has the largest discrepancies. I suggest that the authors make this point clearer by simply stating that the resolved part of the discrepancies is mainly associated with fv.

We agree that it should be clarified that the largest resolved discrepancy is associated with fv but that unresolved discrepancies, found in R, can also be large. We modify the abstract and conclusion accordingly.

I am concerned with their definition of high-agreement and low-agreement SSW events. Those events were defined by the standard deviation of the Coriolis torque averaged from 45-85N. fv is not even an effective measure for SSW events. There are times (within ~5-15 day average window) when zonal mean fv is large but there is no SSW. I do not think that it is appropriate to define the strength of a SSW event (i.e the strongest or weakest) or the associated discrepancies just by using fv. Again, this is because both SSW and the changes in fv and their associated uncertainties are consequences of wave mean-flow interaction.

Although the Coriolis torque produced by the meridional circulation is not a common measure for SSW events, our analysis is motivated by the fact that it is a major source of uncertainty of the momentum budget, and one of the major circulation changes that lead to SSW events. We did not state anywhere that we judged the strength of SSW events using fv. We noted that events with largest discrepancies in fv had a more intense deceleration of zonal-mean zonal wind. As the reviewer suggests, discrepancies in fv can result from discrepancies in wave drag, but also from biases in the mean state and data assimilation procedure (Kobayashi and Iwasaki, 2016; Uppala et al., 2005).

We show in supplementary Fig. 1 the outcome of classifying high agreement SSWs and low agreement SSWs using EP flux divergence, a measure of resolved wave drag, instead of the Coriolis torque. Similar to classifying events with the Coriolis torque, events that have larger discrepancies in wave drag show a stronger deceleration and stronger forcings by the Coriolis torque and momentum flux convergence, as well as stronger forcing for deceleration by EPFD. Since our focus was not on the terms of the transformed Eulerian mean momentum equation, we do not discuss of this analysis in the manuscript.

[Figure]

**Supplementary Figure 1: Similar to Fig. 11 of the manuscript but for comparing SSWs with small (HASSWs – dashed lines) and large uncertainties (LASSWs – solid lines) based on EPFD instead of the Coriolis torque.**

Other than the above points, the paper is well written in general. I suggest publication with some effort to improve the clarity of the expressions. More specific comments are provided below.

Specific comments:

1) Line 15, page 1: "the onset of SSW events, a period characterized by unusually large fluxes of planetary-scale waves from the troposphere to the stratosphere". This sentence holds true only if the period is ~40

days (Polvani and Waugh 2004). The correction between the wave fluxes (or v'T') would become much reduced if the averaging period is only 5-15 days, which is used in this study (i.e. figures 7-8 and figures 11-12). At these shorter time scales, stratospheric internal variation becomes important. This is precisely why the models cannot predict the timing or the initialization of SSWs. The authors must be careful when they discuss their results and when they related to the EP flux divergence to those from the troposphere.

Whereas Polvani and Waugh, (2004) noted that wave fluxes integrated over a period of 40 days were well correlated with the strength of the stratospheric polar vortex, other studies reported a strong link between short-lived bursts of planetary-scale wave activity and the rapid deceleration of the stratospheric polar vortex (Martineau and Son, 2015; McDaniel and Black, 2005; Sjoberg and Birner, 2014). We agree with the reviewer that the intrinsic variability of the stratosphere is important as it can significantly influence the amount of wave activity that can propagate from the troposphere but without a source of wave activity within the troposphere, vortex vacillations would not occur. Since the sentence the reviewer is referring to is supported by our results and previous studies, we decide to keep it as is in the revised manuscript.

2) Line 20, page 1: "The strongest SSWs being subject to larger discrepancies among reanalyses". This sentence gives one impression that there is an accepted definition of "the strongest SSWs". Naturally, the readers would think that these events produced the warmest temperature or strongest easterly winds. Is this true?

We agree that there is no commonly accepted definition of what is a strong SSW. We thus clarify that SSWs with the most intense deceleration of zonal-mean zonal wind show larger discrepancies among reanalysis data sets.

3) Line 15-16, page 3. It is better to state that the previous assessment was mainly for the extratropics. In the tropics where the QBO becomes important, higher vertical and horizontal resolution should lead to much improved dynamical consistency.

We clarify that this assessment was done in the extratropics. We agree that repeating the analysis in the tropical region may lead to different conclusions.

4) Line 21-22, page 4. The last term R also accounts for non-conservative processes, such as Rossby wave breaking (RWB). During SSW, planetary-scale RWB can play an important role. Interestingly, the largest error is associated with R rather than fv term.

The process of planetary-scale wave breaking (not small-scale gravity wave breaking) is largely conservative and resolved by reanalyses until wave activity is transferred to physical scales near the limit of what the model can resolve. Numerical diffusion, that we already mentioned in the manuscript, then dissipates wave activity at these small scales. This diffusion is included in R.

5) Lines 12-16, page 7. Now I understand that the definition is based on the largest discrepancies in the Coriolis torque. This needs to be made clearer in the abstract when you mentioned the strongest SSWs because there is no such a definition in terms of the known or accepted description of the SSWs. Also, see my general comments for further concerns.

We now clarify that the uncertainties in the Coriolis torque are larger when SSW events display a more intense deceleration of the stratospheric polar vortex.

6) Line 2, page 8. "The evolution of geopotential height contours". Please include the values here (not just in the figure caption) and justify why those values are used to describe the polar vortex. Ertel Potential vorticity should be a much better quantity for this purpose and why not to use EPV?

We now mention the values of these contours in the text. The reason we chose these contours is simply because they clearly illustrate the shape of the stratospheric polar vortex throughout the life cycle of the 2009 SSW event and we now mention it in the manuscript. We agree that EPV is a better quantity to study the dynamical evolution of SSWs as it is conserved for conservative flows. However, we wished to use a less derived quantity to illustrate the shape of the stratospheric polar vortex. Geopotential height serves our purpose well as it is parallel to the geostrophic flow. Geopotential height is used frequently to describe the evolution of SSW events (Charlton and Polvani, 2007; Limpasuvan et al., 2004; Martineau and Son, 2015; Seviour et al., 2013).

7) Figures 3-4. It is really hard to qualify the spread or discrepancies based on the color bar used.

We are now using a better-suited colormap.

8) Line 7, page 15. "Terms that are left of the QG from of the momentum equation provide much smaller forcing for zonal wind tendency during SSW events . . . Their differences from one reanalysis ". I disagree for the following reasons. 1). The two QBO terms are of the opposite sign in general (see figure 7). If they are added together, the sum would have a comparable magnitude when it is compared with the other terms. 2). It is well-known that the SSW events often involve breaking of finite amplitude waves.

Such an effect cannot be accounted for by 2.5 resolution pressure level data. Please reword the part to avoid the possibility of misleading the readers. See my general comments for further information.

Even when the two QG terms, which are often opposed, are added together, their total contribution is usually larger than the non-QG terms and explain the largest fraction of zonal wind tendencies (see supplementary Fig. 3). But, we agree with you that it may be more accurate to tone down our affirmation by replacing *much smaller* by *smaller*.

Most of the deceleration during SSW events results from a large amplification of planetary-scale waves in the stratosphere (Martineau and Son, 2015; Solomon, 2014). The deceleration of zonal wind is strongest when the QGPV field is deformed by wave-1 or wave-2 disturbances (v'q'). When the waves reach large amplitudes, wave-breaking will indeed redistribute wave activity to smaller scales (this does not necessarily lead to meridional QGPV fluxes and deceleration of zonal-mean zonal wind). As mentioned earlier, once wave activity is transferred to smaller scales, it will be dissipated by numerical diffusion and be included in the residual term. Overall, most of the deceleration of zonal wind during SSW event is well accounted for by the QG terms in reanalyses (Martineau and Son, 2015).

9) I am not sure whether or not figures 8 and 9 is needed. Would it be more concise or informative if the figures were combined as one and show the two groups: the latest versus older generation reanalysis products?

We decide to keep these figures since they may be useful for reanalysis centers and reanalysis users to evaluate discrepancies of specific reanalyses with respect to others and locate the regions of the atmosphere responsible for these biases. This would not be possible by showing only composites of newer versus older reanalyses.

10) Lines 18-28, Page 22. I suggest that the authors to check would the same spread or results be obtained using the residual term R and its standard deviation to define HASSWs and LASSWs. Same applies to figures 11 and 12.

As suggested by the reviewer, we verify if the same result would hold by defining HASSWs and LASSWs using the residual of the momentum budget instead of the Coriolis torque. The results are shown here with supplementary Fig. 2. Events with large discrepancies of the residual show a more intense deceleration, although the forcings by the Coriolis torque and momentum fluxes are more similar compared to the differences between HASSWs and LASSWs defined with the Coriolis torque. This suggests that what differentiates these two categories of events is most likely the strength of unresolved forcing included in R, such as gravity wave drag. Since unresolved forcings are not the foci of this work, we elect to not discuss of this result further in the manuscript.

[Figure]

**Supplementary Figure 2: Similar to Fig. 11 of the manuscript but for comparing SSWs with small (HASSWs – dashed lines) and large uncertainties (LASSWs – solid lines) based on R instead of the Coriolis torque.**

11) Line 14, page 26. See general comments. The results do not suggest that the discrepancies in those non-QG terms are smaller than the QG-terms.

We show in supplementary Fig. 3 that the sum of QG terms is typically larger than the sum of non-QG terms. We agree that the difference may not be as large as we suggest and thus will tone down our statements wherever applicable.

[Figure]

**Supplementary Figure 3: Same as Fig. 5 of the manuscript except that the QG and non-QG terms are summed together.**

12) Line 23, page 26. "Most of the residual in the stratosphere is correlated to uncertainties in the Coriolis torque". This is very interesting and somehow expected. My explanation is as follows. In the upper stratosphere, gravity wave breaking and finite amplitude wave activities appear regularly there but their propagation cannot be well captured by 2.5 degree pressure level data. Their effects on the polar vortex or zonal mean zonal wind would be included in R or the vertical momentum flux term especially when the QG-terms are calculated by using variables such as u and v, as it is done by this study. On the other hand, when the EP flux divergence is included as in the transformed Eulerian mean equations, the variation of wave forcing would be better resolved by the data used. This is because Del F accounts for the vertically propagating wave not just the meridionally propagating waves. This is confirmed by figure 11. The figures shows, at 3 hPa, the temporal evolution of the zonal mean wind tendency follows better with the EP flux

divergence, less so in terms of fv. Thus, I would think that it is the uncertainties associated with non-resolved wave forcing caused the spread in fv, rather than the other way around.

The bulk of the deceleration of zonal-mean zonal wind in the stratosphere is produced by planetary-scale waves and is thus well resolved by 2.5 degree pressure level data (most of meridional fluxes of QGPV, and thus EP flux convergence, are resulting from large scale motion by wave-1 and wave-2 planetary-scale waves). We agree however that wave-breaking may lead to fluxes at scales smaller than what can be resolved by a 2.5 degree grid and that the effect of gravity waves, which is not included in our diagnostic, will show up in R.

We agree that the spread in fv may result from discrepancies in wave drag, either from planetary waves or gravity waves and will make sure to clarify this point in the revised manuscript. Reanalysis datasets, however, are not only following their own physics, they assimilate data and are also susceptible to discrepancies in data assimilation (Kobayashi and Iwasaki, 2016; Kobayashi et al., 2015; Uppala et al., 2005). We make modifications to the conclusion to take into account these contributions.

Minor comments:

1) Line 29, page 1. Too many citations here for motivation.

We reduce the number of citations.

2)Line 5, page 2. Two daughter vortices -> two vortices.

Corrected

3) Line 6, page 2. Please be more specific about the differences. Otherwise, delete the sentence as it adds no information.

We remove this sentence.

4) Line 9, page 2. "the general signature". What is it? Please be more

specific.

We now clarify what is this signature by mentioning the deceleration of zonal-mean zonal wind and the warming of the polar cap.

5) Line 25, ., -> ,

Corrected

6) Line 16, high stratosphere -> upper stratosphere.

Corrected

Charlton, A. J. and Polvani, L. M.: A New Look at Stratospheric Sudden Warmings. Part I: Climatology and Modeling Benchmarks, J. Clim., 20(3), 449–469, doi:10.1175/JCLI3996.1, 2007.

Kobayashi, C. and Iwasaki, T.: Brewer-Dobson circulation diagnosed from JRA-55, J. Geophys. Res. Atmos., 121(4), 1493–1510, doi:10.1002/2015JD023476, 2016.

Kobayashi, S., Ota, Y., Harada, Y., Ebita, A., Moriya, M., Onoda, H., Onogi, K., Kamahori, H., Kobayashi, C., Endo, H., Miyaoka, K. and Takahashi, K.: The JRA-55 Reanalysis: General Specifications and Basic Characteristics, J. Meteorol. Soc. Japan. Ser. II, 93(1), 5–48, doi:10.2151/jmsj.2015-001, 2015.

Limpasuvan, V., Thompson, D. W. J. and Hartmann, D. L.: The Life Cycle of the Northern Hemisphere Sudden Stratospheric Warmings, J. Clim., 17(13), 2584–2596, doi:10.1175/1520-0442(2004)017<2584:TLCOTN>2.0.CO;2, 2004.

Martineau, P. and Son, S.-W.: Onset of Circulation Anomalies during Stratospheric Vortex Weakening Events: The Role of Planetary-Scale Waves, J. Clim., 28(18), 7347–7370, doi:10.1175/JCLI-D-14-00478.1, 2015.

McDaniel, B. A. and Black, R. X.: Intraseasonal Dynamical Evolution of the Northern Annular Mode, J. Clim., 18(18), 3820–3839, doi:10.1175/JCLI3467.1, 2005.

Polvani, L. M. and Waugh, D. W.: Upward Wave Activity Flux as a Precursor to Extreme Stratospheric Events and Subsequent Anomalous Surface Weather Regimes, J. Clim., 17(18), 3548–3554, doi:10.1175/1520-0442(2004)017<3548:UWAFAA>2.0.CO;2, 2004.

Seviour, W. J. M., Mitchell, D. M. and Gray, L. J.: A practical method to identify displaced and split stratospheric polar vortex events, Geophys. Res. Lett., 40(2), 5268–5273, doi:10.1002/grl.50927, 2013.

Sjoberg, J. P. and Birner, T.: Stratospheric wave-mean flow feedbacks and sudden stratospheric warmings in a simple model forced by upward wave-activity flux, J. Atmos. Sci., (1976), 141006071034003, doi:10.1175/JAS-D-14-0113.1, 2014.

Solomon, A.: Wave Activity Events and the Variability of the Stratospheric Polar Vortex, J. Clim., 27(20), 7796–7806, doi:10.1175/JCLI-D-13-00756.1, 2014.

Uppala, S. M., Kallberg, P. W., Simmons, A. J., Andrae, U., Bechtold, V. D. C., Fiorino, M., Gibson, J. K., Haseler, J., Hernandez, A., Kelly, G. A., Li, X., Onogi, K., Saarinen, S., Sokka, N., Allan, R. P., Andersson, E., Arpe, K., Balmaseda, M. A., Beljaars, A. C. M., Berg, L. Van De, Bidlot, J., Bormann, N., Caires, S., Chevallier, F., Dethof, A., Dragosavac, M., Fisher, M., Fuentes, M., Hagemann, S., Holm, E., Hoskins, B. J., Isaken, L., Janssen, P. A. E. M., Jenne, R., McNally, A. P., Mahfouf, J.-F., Morcrette, J.-J., Rayner, N. A., Saunders, R. W., Simon, P., Sterl, A., Trenberth, K. E., Untch, A., Vasiljevic, D., Viterbo, P. and Woollen, J.: The ERA-40 re-analysis, Q. J. R. Meteorol. Soc., 131, 2961–3012, 2005.

[revised manuscript text omitted]

SSWs are also known to present a large diversity in terms of how the stratospheric polar vortex is distorted in the course of the events. While some events occur due to a displacement of the vortex, others occur from a splitting (Charlton and Polvani, 2007). These two types of SSW events result from different planetary-scale wave forcing in the stratosphere and can affect the tropospheric flow in different ways (Bancalá et al., 2012; Lehtonen and Karpechko, 2016; Martineau and Son, 2015; Mitchell et al., 2013; Smith and Kushner, 2012). It is thus possible that one type or the other is subject to larger uncertainties in reanalysis datasets. To test whether this is the case, the two types of SSWs, i.e., split SSWs (SSWS) and displacement SSWs (SSWD), are classified using vortex moment diagnostics described in Seviour et al. (2013) (see Table 2). (Bancalá et al., (2012), however, pointed out that some

wavenumber-2 SSW events were not purely forced by fluxes of wavenumber-2 wave activity but that wavenumber-1 fluxes also contributed to the preconditioning of these events. We therefore also classify events according to the dominant forcing prior to the onset date. To this end, we compute the ratio of wavenumber-1 to wavenumber-2 vertical EP flux at 100 hPa over 14 days preceding the onset date. The eight events that have the largest ratio are classified as W1-dominant event and the eight events with the smallest ratio are classified as W2-dominant events. The outcome of this classification is indicated in Table 2.

SSWs are further classified based on whether the reanalysis datasets included in the LRE agree or disagree for a specific diagnostic. Since the Coriolis torque is the forcing term that shows the largest discrepancies among reanalysis datasets in the stratosphere (Martineau et al., 2016), it is used here to quantify the level of agreement. The standard deviation of the Coriolis torque averaged from 45°N to 85°N among the LRE is considered each day from 10 days before the central date to 5 days after the central date which is shown later to be the period when the Coriolis force is largest during SSW events. The agreement is then defined for each event as the maximum standard deviation observed during the period considered. Events are finally classified into two categories: high-agreement SSWs (HASSWs) or low-agreement SSWs (LASSWs) whether they are in the lower 33.3% or upper 33.3% of the agreement index of all SSWs. The type of each event is indicated in Table 2.

**3 Vortex geometry during the 2009 SSW**

Since the geometry of the stratospheric vortex may vary widely from one event to another, the structure of any particular event can be heavily obscured when performing composite means. We therefore first proceed to illustrate and compare the morphology of the stratospheric vortex among reanalysis datasets for a representative SSW event that is well documented in the literature. We choose the January 2009 SSW, an event characterized by a vortex splitting that resulted from an unusually large amount of upward EP flux by wavenumber two (Harada et al., 2010; Manney et al., 2009). Reanalysis datasets show qualitatively similar downward coupling during this event (Martineau and Son, 2010). Figure 2 illustrates the evolution of the 30 km and 38.5 km geopotential height contours at 10 and 3 hPa, respectively. These contours are chosen because they clearly illustrate the distortion of the stratospheric polar vortex during the life cycle of the 2009 SSW. 
[revised manuscript text omitted]
. On the other hand, when considering the dominant fluxes of wave activity producing SSW evens, we find that out of 7 LASSWs, four are W1-dominant and 1 is W2 dominant and out of 7 HASSWs, 1 is W1-dominant and 3 are W2-dominant. This seems to indicate that wavenumber-1 wave drag is responsible for larger uncertainties in reanalysis datasets but a detailed analysis reveals that inter-reanalysis spread is not markedly different between W1-dominant and W2-dominant events (not shown) suggesting that it is the intensity of wave drag rather than the longitudinal scale of wave activity that is linked to uncertainties among reanalyses.

[revised manuscript text omitted]

skill of tropospheric forecasts, Q. J. R. Meteorol. Soc., 141(689), 987–1003, doi:10.1002/qj.2432, 2015.

Uppala, S. M., Kallberg, P. W., Simmons, A. J., Andrae, U., Bechtold, V. D. C., Fiorino, M., Gibson, J. K., Haseler, J., Hernandez, A., Kelly, G. A., Li, X., Onogi, K., Saarinen, S., Sokka, N., Allan, R. P., Andersson, E., Arpe, K., Balmaseda, M. A., Beljaars, A. C. M., Berg, L. Van De, Bidlot, J., Bormann, N., Caires, S., Chevallier, F., Dethof, A., Dragosavac, M., Fisher, M., Fuentes, M., Hagemann, S., Holm, E., Hoskins, B. J., Isaken, L., Janssen, P. A. E. M., Jenne, R., McNally, A. P., Mahfouf, J.-F., Morcrette, J.-J., Rayner, N. A., Saunders, R. W., Simon, P., Sterl, A., Trenberth, K. E., Untch, A., Vasiljevic, D., Viterbo, P. and Woollen, J.: The ERA-40 re-analysis, Q. J. R. Meteorol. Soc., 131, 2961–3012, 2005.